# Boosting the Photoreactivity of g-C$_3$N$_4$ towards CO$_2$ Reduction by Polymerization of Dicyandiamide in Ammonium Chloride

Zhi Wang [1], Shixin Chang [1], Mengxue Yu [1], Zaiwang Zhao [2,3,*], Qin Li [1] and Kangle Lv [1,*]

1 College of Resources and Environment, South-Central Minzu University, Wuhan 430074, China; wangzhi0030@163.com (Z.W.); csx1335243806@163.com (S.C.); yumengxue1104@163.com (M.Y.); liqin0518@mail.scuec.edu.cn (Q.L.)
2 College of Energy Materials and Chemistry, Inner Mongolia University, Hohhot 010070, China
3 College of Chemistry and Chemical Engineering, Inner Mongolia University, Hohhot 010070, China
* Correspondence: zwzhao@imu.edu.cn (Z.Z.); lvkangle@mail.scuec.edu.cn (K.L.)

**Abstract:** As a typical organic semiconductor photocatalyst, graphitic carbon nitride (g-C$_3$N$_4$) suffers from low photocatalytic activity. In this paper, g-C$_3$N$_4$ was prepared by polymerization of dicyandiamide (C$_2$H$_4$N$_4$) in the presence of ammonium chloride (NH$_4$Cl). It was found that the addition of ammonium chloride can greatly improve the photocatalytic activity of g-C$_3$N$_4$ towards CO$_2$ reduction. The optimal photocatalyst (CN-Cl 20) exhibited a CO$_2$-to-CO conversion activity of 50.6 $\mu$molg$^{-1}$h$^{-1}$, which is 3.1 times that of pristine bulk g-C$_3$N$_4$ (BCN) that was prepared in the absence of any ammonium chloride. The enhanced photoactivity of g-C$_3$N$_4$ was attributed to the combined effects of chloride modification and an enlarged specific surface area. Chloride modification of g-C$_3$N$_4$ can not only reduce the bandgap, but also causes a negatively shifted conduction band (CB) potential level, while ammonia (NH$_3$) gas from the decomposition of NH$_4$Cl can act as a gas template to exfoliate layered structure g-C$_3$N$_4$, improving the specific surface from 6.8 to 21.3 m$^2$g$^{-1}$. This study provides new ideas for the synthesis of highly efficient g-C$_3$N$_4$-based photocatalytic materials for CO$_2$ conversion and utilization.

**Keywords:** g-C$_3$N$_4$; photocatalysis; CO$_2$ reduction; ammonium chloride





## 1. Introduction

With the rapid development of the economy, the consumption of fossil fuels has caused massive emissions of CO$_2$, resulting in a serious environmental pollution and ecology crisis. The photocatalytic CO$_2$ reduction reaction (CO$_2$RR) has provided a sustainable way to solve the problem, as it is considered as a potential method with the goal of accomplishing carbon neutrality and carbon compliance. Although the production of solar fuels (methanol, ethanol, ethylene, etc.) [1] by the CO$_2$RR still remains a challenge in terms of selectivity and efficiency, it can be obtained that the reduction of CO$_2$ to CO involves only two electron transfers and two proton transfers. Therefore, the CO$_2$RR is currently the most likely pathway for efficient experimental CO$_2$ reduction [2].

Two-dimensional (2D) layered materials have gained a lot of attention due to their various applications in energy generation, energy storage, and catalysis. Among them, graphitic carbon nitride (g-C$_3$N$_4$), as a typical 2D layered photocatalyst, possesses the merits of non-toxicity, facile preparation, and suitable bandgap [3]. Nevertheless, the photocatalytic performance of g-C$_3$N$_4$ is limited by the charge carriers, because of the slow charge mobility and fast complexation. To compensate for these drawbacks, scientists have made various attempts, such as building various nanostructures [4], surface modification [5], constructing heterojunctions [6], and doping with chemical elements [7]. Among the above strategies, surface modification has proved to be an effective way to tune the electronic structure of g-C$_3$N$_4$, achieve modulation of the bandgap, and enhance the photocatalytic activity [8].

The introduction of the gas-phase template method in the preparation of carbon nitride can effectively simplify the experimental process and reduce the experimental cost [9]. The pore structure generated by the reaction with the precursor enhances the porosity of the material due to the generation of a large number of bubbles during the reaction process. At the same time, the pore structure formed by the escape of gas can play the role of material transfer and electron–hole separation, which ultimately improves the performance of the catalyst. The use of $NH_4Cl$ as a gas templating agent does not require to remove the template with chemical reagents in the subsequent process, which significantly increases the specific surface area and creates more active sites without changing the basic structure of g-$C_3N_4$ [10].

Since the photocatalytic activity of carbon nitride was discovered, the preparation and modification of carbon nitride have been studied, such as molten salt methods [11], solvothermal methods [12], and solid salt methods [13]. Meanwhile, acid mist and other thermal polymerization methods have received attention because of their ability to create a suitable reaction environment during the synthesis process. Ammonium chloride starts to decompose at 100 °C and completely decomposes to ammonia and hydrochloric acid at 337 °C [14]. In the process of carbon nitride synthesis, the precursor dicyandiamide and ammonium chloride are mixed at first. During calcination, a large amount of ammonia affects the polymerization of carbon nitride to increase its specific surface area, which is considered the gas-phase template method [15]. However, in previous studies, the calcination of carbon nitride mixed with a small amount of ammonium chloride did not significantly improve the photoelectrochemical properties of the modified carbon nitride, nor did it have doping of the Cl element. The improvement in performance is due to the increase of the specific surface area by the gas-phase template method [16]. In our experiments, we found that when the ratio of dicyandiamide and ammonium chloride was 1:20, a large amount of ammonia and hydrochloric acid produced by the decomposition of the ammonium chloride formed an acid mist. The carbon nitride was polymerized in the environment of the acid mist, sufficiently modifying it. The doping of Cl altered the charge distribution on the surface of the carbon nitride, which improved the efficiency of the carrier transport.

In this work, we used an acid mist thermal polymerization approach to prepare porous graphitic carbon nitride. This porous g-$C_3N_4$ exhibits a high specific surface area and excellent photocatalytic activity towards $CO_2$ reduction. A series of characterizations were employed to investigate the effects of Cl modification on the crystal phase, morphology, specific surface area, band alignment, as well as the photoreduction $CO_2$ performance of g-$C_3N_4$. The CN-Cl 20 sample shows the highest photocatalytic activity towards $CO_2$ reduction, which is ascribed to its high specific surface area, narrow bandgap, negatively shifted CB position, and fast interfacial charge transfer rate.

## 2. Results and Discussion

### 2.1. Effect of Ammonium Chloride on the Morphology of CN

According to the literature, ammonium chloride usually starts to decompose and produce $NH_3$ and HCl at elevated temperatures (338 °C) [17]. Consequently, $NH_4Cl$ was employed as the gas template in our experiment. The preparation process of g-$C_3N_4$ catalysts is illustrated in Figure 1. On the one hand, onsite gas production is beneficial to exfoliate bulk g-$C_3N_4$, and thus result in an enlarged specific surface area. On the other hand, the acid mist environment (ammonia and hydrochloride acid vapor) is favorable for the thermal polymerization of the $C_2H_4N_4$ precursor. This was confirmed from the increased yields (Table S1) of BCN (43%) to CN-Cl 20 (47%).

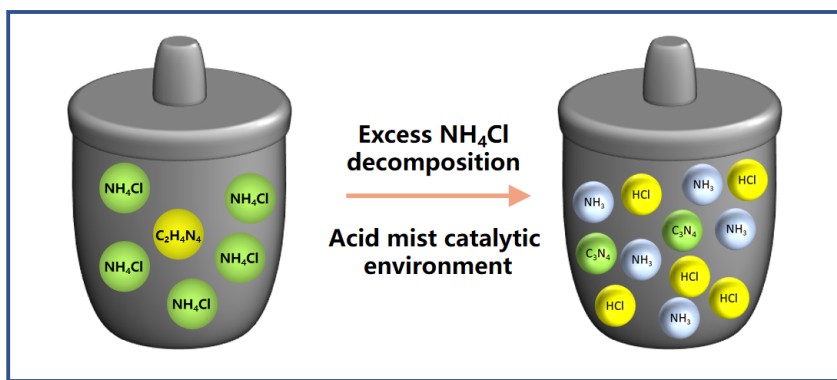

**Figure 1.** The scheme showing the polymerization process of CN in an acid mist environment.

The morphologies of BCN and CN-Cl 20 samples are observed in Figure 2a,b. Compared to the smooth surface of BCN, the CN-Cl 20 sample displays a remarkably multi-porous structure, suggesting an enlarged specific surface area. The different morphologies of the BCN and CN-Cl 20 samples are in line with their TEM images (Figure 2c,d). Besides, the pore diameter of the CN-Cl 20 sample, observed from its high-resolution TEM image, is about 17 nm (Figure 2e). C, N, and Cl elements of the CN-Cl 20 catalyst are detected in its element mapping (Figure 2f), confirming the successful introduction of Cl into g-$C_3N_4$.

### 2.2. Effect of Ammonium Chloride on the Microstructures of CN

The Brunauer–Emmett–Teller specific surface area ($S_{BET}$) of the photocatalyst is closely dependent on its $CO_2$ adsorption and reaction active sites. The measured nitrogen adsorption–desorption isotherms and corresponding pore size distribution curves of the as-prepared samples are shown in Figure 3a. All catalysts exhibit typical IV-type adsorption isotherms and $H_3$-type hysteresis loops in the relative pressure of 0.8~1.0, indicating the presence of slit pores in the stacked nanosheets [18]. The BET surface area, the pore volume, and the pore size of the as-prepared BCN and CN-Cl x samples are summarized in Table 1. In comparison with BCN (6.8 m$^2$ g$^{-1}$), the CN-Cl x samples have much larger $S_{BET}$ values. Among the CN-Cl x samples, CN-Cl 2 displays the largest $S_{BET}$ (30.7 m$^{-2}$ g$^{-1}$), which is 4.5 times higher than that of BCN. When the amount of NH$_4$Cl increases to 20 g, the $S_{BET}$ of the obtained sample (CN-Cl 5) reduces to 21.3 m$^2$ g$^{-1}$. A further increase in the amount of NH$_4$Cl has little effect on the specific surface area of the CN-Cl x samples (see $S_{BET}$ of CN-Cl 10 and CN-Cl 20 samples). This phenomenon is possibly due to the collapse of the g-$C_3N_4$ porous structure during the thermal polymerization after the excess use of the NH$_4$Cl template. The pore size distribution curves of all BCN and CN-Cl x samples show similar trends to their specific surface areas (Table 1). According to the pore distributions of the as-prepared CN-Cl x samples, all catalysts display 50–100 nm large pores and ~5 nm mesopores (Figure 3b). In addition, the average particle sizes of BCN and CN-Cl 20 samples are 42.6 nm and 26.9 nm, respectively, which indicates that the use of the NH$_4$Cl gas template hinders the aggregation of g-$C_3N_4$ nanoparticles during calcination.

**Table 1.** BET surface area, pore volume, and pore size of the as-prepared BCN and CN−Cl x samples.

| Sample | BCN | CN-Cl 1 | CN-Cl 2 | CN-Cl 5 | CN-Cl 10 | CN-Cl 20 |
|---|---|---|---|---|---|---|
| $S_{BET}$ (m$^2$ g$^{-1}$) | 6.8 | 10.7 | 30.7 | 21.3 | 20.6 | 21.3 |
| Total pore volume (cm$^3$g$^{-1}$) | 0.07 | 0.09 | 0.29 | 0.20 | 0.13 | 0.16 |
| Average particle size (nm) | 42.6 | 35.2 | 38.1 | 39.2 | 26.6 | 26.9 |

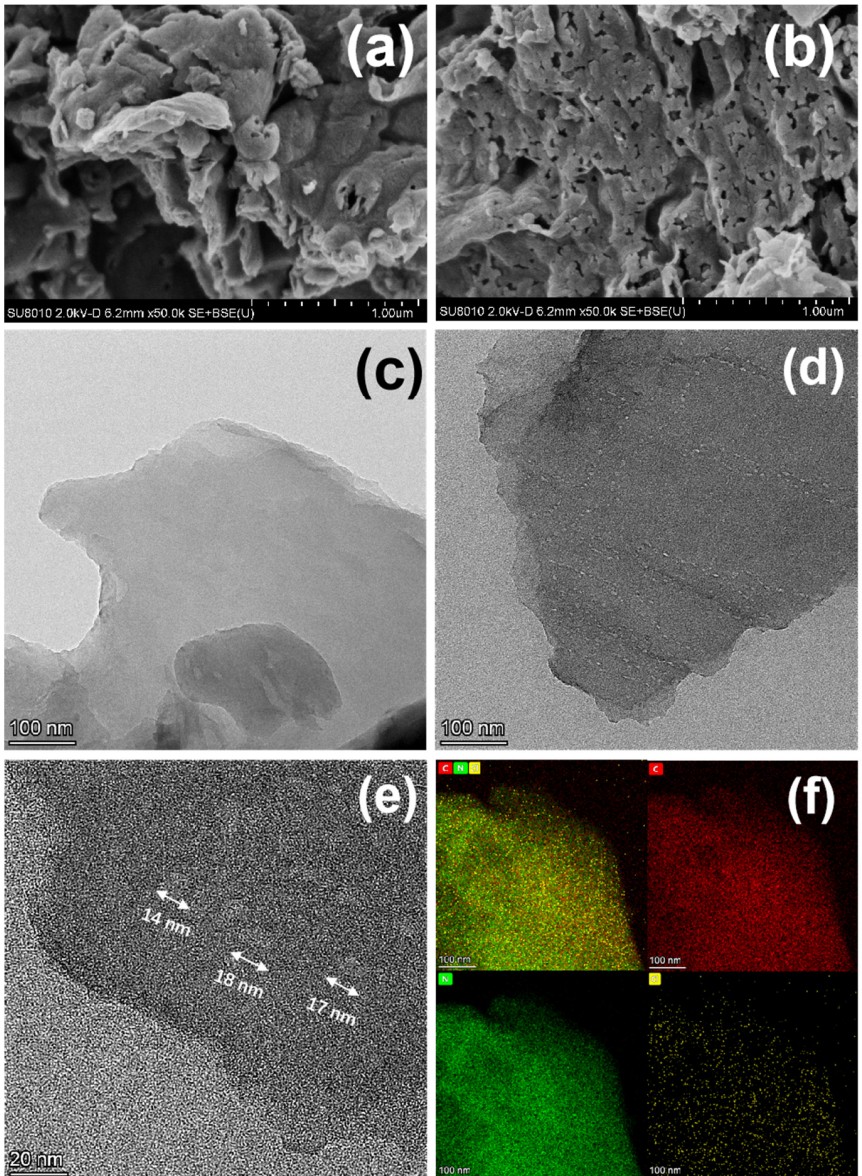

**Figure 2.** SEM images of the BCN (**a**) and CN−Cl 20 photocatalysts (**b**). TEM images of the BCN (**c**) and CN−Cl 20 samples (**d**). HRTEM image of the CN−Cl 20 sample (**e**) and its corresponding HAADF−STEM−EDS mapping images (**f**): C (red), N (blue), and Cl (yellow).

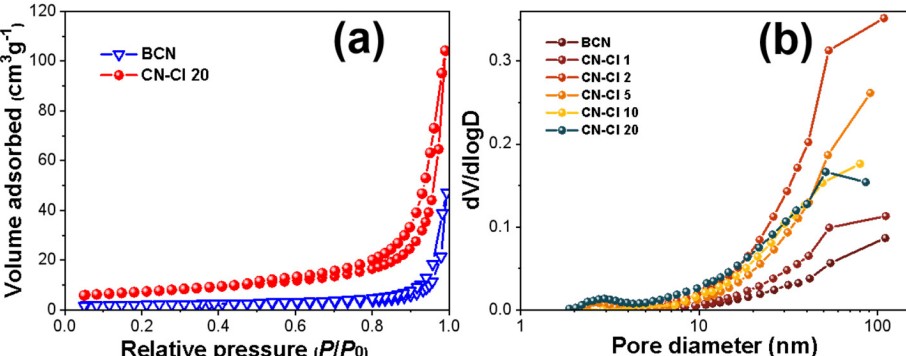

**Figure 3.** N$_2$ adsorption–desorption isotherms (**a**) and the corresponding pore size distribution curves (**b**) of the as–prepared BCN and CN−Cl x samples.

### 2.3. Phase Structures and Surface Chemical States

X-ray diffraction (XRD) was employed to identify the structures of the as-prepared BCN and CN-Cl x samples (Figure 4a). According to the literature, the peaks at 13.7° and 27.8° are attributed to the (100) and (002) planes of g-$C_3N_4$, respectively. The weak diffraction (100) peak stems from the in-plane structural stacking motif of the tri-S-triazine unit, while the strong (002) peak originates from the interplanar stacking reflections of the conjugated aromatic system in g-$C_3N_4$. From Figure S1, it can be clearly seen that a diffraction shift of (002) peaks from 27.8° (BCN) to 28.1° (CN-Cl 20), which implies that the layer spacing of g-$C_3N_4$ slightly increases after introduction of Cl [19]. In addition, by increasing the amount of $NH_4Cl$, the (002) peak intensities of the CN-Cl x samples gradually become weaker, which indicates that Cl was successfully introduced into the g-$C_3N_4$.

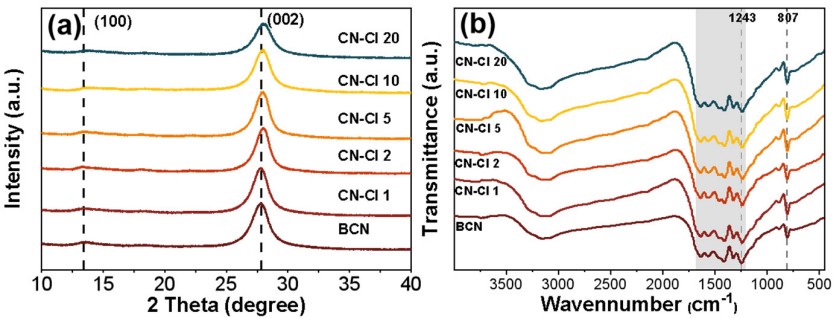

**Figure 4.** XRD pattern (**a**) and FTIR spectra (**b**) of the BCN and CN−Cl x samples.

To investigate the effect of Cl modification on the frame structure of g-$C_3N_4$, FTIR spectra of pristine BCN and Cl-modified CN samples were recorded (Figure 4b). The broad peaks at 3000–3500 $cm^{-1}$ are ascribed to asymmetric and symmetric vibrations of the O-H group in water. The peaks in the range 1200~1680 $cm^{-1}$ belong to the typical stretching vibrational mode of the tri-S-triazine ring in g-$C_3N_4$, while the absorption peak at 807 $cm^{-1}$ is attributed to the bending vibrational mode of the heptazine units. These characteristic peaks of g-$C_3N_4$ remain after Cl modification, which indicates that introduction of Cl does not break the structure of the tri-S-triazine ring. The FTIR peak at 1243 $cm^{-1}$ is assigned to the triazine unit and stretching vibrations of C-N, respectively [20].

The X-ray photoelectron spectra of the as-prepared BCN and CN-Cl 20 samples were collected to analyze doping and surface speciation. As shown in Figure 5a,b, the presence of a Cl 2*p* peak at 197.3 eV is clearly observed in the CN-Cl 20 sample, which implies that Cl was successfully introduced into the sample. The three peaks at 288.3, 286.1, and 284.8 eV in the high-resolution C1*s* spectrum of the CN-Cl 20 sample are attributed to N=C-N, C-$NH_X$, and C-C, respectively (Figure 5c). The high-resolution N1*s* spectrum of the CN-Cl 20 sample showed three peaks at 401.2, 399.9, and 398.8 eV, which are ascribed to C-$NH_X$, N-$(C)_3$, and C=N-C, respectively. In comparison with BCN, both the C1*s* and N1*s* peaks of the CN-Cl 20 sample are slightly shifted, which can be explained by the partial valence state change of C and N after Cl modification.

### 2.4. Photocatalytic Performance

Photoreduction of $CO_2$ was carried out under the irradiation of a simulated visible 300 W Xenon arc lamp with a cut-off filter ($\lambda \geq 420$ nm), in the absence of any sacrificial agent. The products of the photocatalytic $CO_2$ reduction were mainly carbon monoxide (CO) and a small amount of methane ($CH_4$) (see Figure S3). Therefore, the CO production rate was used to evaluate the photocatalytic activity of the CN samples. Figure 6a compares the CO production rates among different photocatalysts for $CO_2$ reduction. Compared to BCN, the CN-Cl x samples exhibit a pronouncedly higher photocatalytic $CO_2$-to-CO conversion efficiency. The photocatalytic CO production rate of CN-Cl 20 is as high as

$50.6 \ \mu mol \ g^{-1} \ h^{-1}$, which is 3 times higher than that of BCN with a CO production rate of only $16.5 \ \mu mol \ g^{-1} \ h^{-1}$ (Figure 6a). After five consecutive cycles, 90% of the photocatalytic performance of the CN-Cl 20 sample remains (Figure 6b).

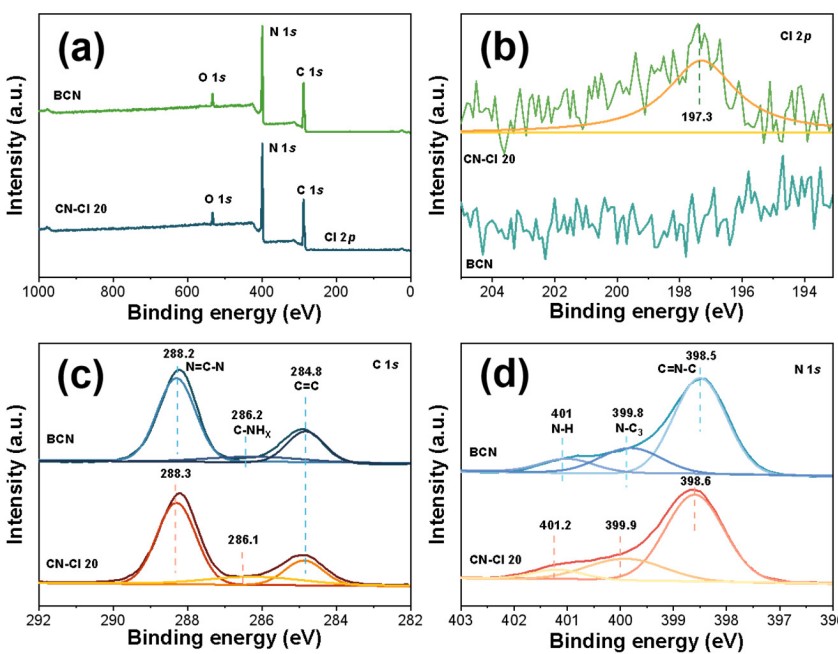

**Figure 5.** XPS survey spectra (**a**) of the CN−Cl 20 sample and its high–resolution XPS spectra of Cl 2$p$ (**b**), C1$s$ (**c**), and N1$s$ (**d**).

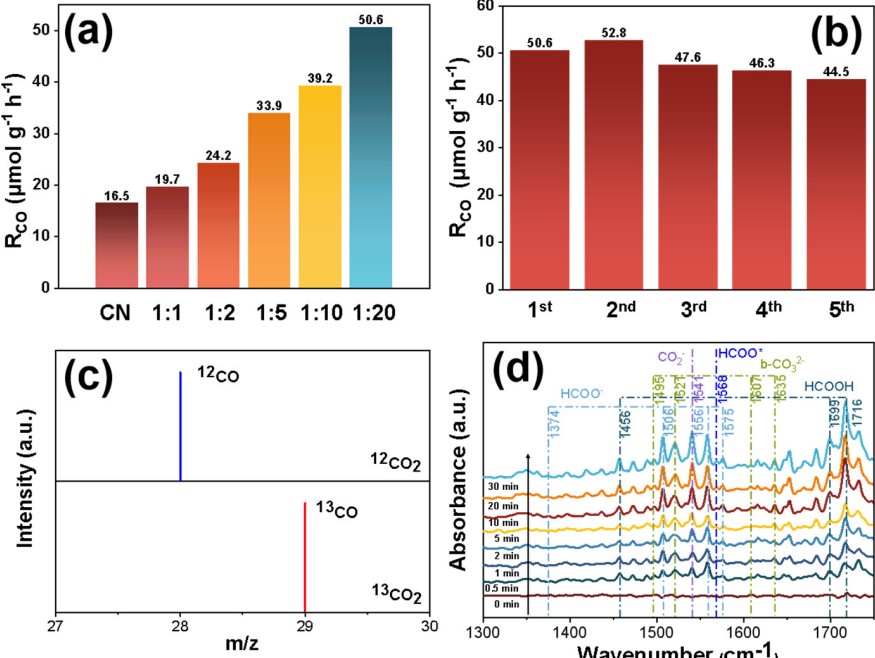

**Figure 6.** Photocatalytic $CO_2$ reduction rates of the BCN and CN−Cl x samples (**a**). Cycling tests of the CN−Cl 20 sample (**b**). MS analysis of photoreduction $CO_2$ productions over CN−Cl 20 using $^{12}CO_2$ and $^{13}CO_2$ as the carbon sources (**c**). In situ DRIFTs spectra of the CN−Cl 20 sample in the dark and under illumination (**d**).

To confirm the conversion process of photocatalytic $CO_2$ to CO, we analyzed the products of the BCN and CN-Cl 20 samples using GC. The experiments demonstrated that no product was observed in the absence of light irradiation, $CO_2$, or photocatalyst. This

result suggests that the light excitation and $CO_2$ are required during this photocatalytic reaction. The $^{13}CO_2$ isotope was employed to confirm the origination of the CO product. As shown in Figure 6c, $^{12}CO$ ($m/z$ = 28) was detected when $^{12}CO_2$ was used as the carbon source, while only the $^{13}CO$ ($m/z$ = 29) line was observed when $^{13}CO_2$ was used as the carbon source. This experiment demonstrated that the CO product of $CO_2$ photoreduction stems from $CO_2$, rather than other carbon sources or impurities [21].

To investigate the photocatalytic process of $CO_2$ reduction, in situ diffuse reflectance Fourier transform infrared spectroscopy (DRIFTs) was used to monitor the reaction intermediates of the $CO_2$ to CO conversion in the photocatalytic reduction process of CN-Cl 20. As shown in Figure 6d, in the absence of $CO_2$ and $H_2O$ vapor, no FTIR peaks were initially detected. When $CO_2$ gas and $H_2O$ vapor were flowing, numerous peaks could be observed. The characteristic peaks located at 1495, 1521, 1607, and 1635 cm$^{-1}$ are attributed to bidentate carbonate (b-$CO_3^{2-}$) [22]. In addition, the peak at 1541 cm$^{-1}$ belongs to monodentate carbonate (m-$CO_3^{2-}$) [23]. These carbonates were derived from $CO_2$ dissolved in water. Under visible light irradiation, various peaks appeared. The peaks centered at 1374, 1506, 1556, and 1575 cm$^{-1}$ are ascribed to formate (HCOO$^-$), while the peaks at 1456, 1699, and 1716 cm$^{-1}$ are attributed to formic acid (HCOOH) [24]. The peak at 1568 cm$^{-1}$ originates from the absorbed COOH* species. To conclude, HCOO$^-$, HCOOH, and COOH* were considered as key intermediates in the photoreduction $CO_2$ to CO reaction. The detailed reaction pathway is deduced as the following Equations (1)–(4). Firstly, the $CO_2$ molecule is adsorbed on the surface of the CN-Cl 20 sample. After that, the absorbed $CO_2$ molecule is reduced to an absorbed HCOOH species by one electron after coupling with a proton (Equation (2)). The resulting COOH speciation is further reduced to an absorbed CO molecule by a photogenerated electron (Equation (3)) [25]. Eventually, the desorption of the absorbed CO molecule occurs (Equation (4)).

$$CN + h\nu \rightarrow e^- + h^+ \tag{1}$$

$$CO_2 + e^- + H^+ \rightarrow COOH^* \tag{2}$$

$$COOH^* + e^- + H^+ \rightarrow CO^* + H_2O \tag{3}$$

$$CO^* \rightarrow CO \uparrow \tag{4}$$

*2.5. Photoelectrochemical Properties*

UV-vis diffuse reflectance spectroscopy (DRS) was used to determine the optical absorption ability of the catalyst. As shown in Figure 7a, all CN-Cl x samples show similar light absorption spectra. The intrinsic absorption band in the region of 200–450 nm is generally considered as $\pi \rightarrow \pi^*$ electron transition of the conjugated aromatic system. The absorption edge of the CN-Cl x samples is slightly red-shifted, which could be attributed to a change in the local electric field on the conjugated aromatic system [26]. The relationship between the Kubellka–Munk function and photon energy is shown in Figure 7b, whereby the bandgap narrowed from 2.82 eV to 2.76 eV with the increase of $NH_4Cl$. The bandgaps of BCN, CN-Cl 1, CN-Cl 2, CN-Cl 5, CN-Cl 10, and CN-Cl 20 were determined as 2.82 eV, 2.80 eV, 2.80 eV, 2.78 eV, 2.77 eV, and 2.76 eV, respectively. The reduced bandgap width means more light can be used for photocatalysis, which is beneficial for the photocatalytic $CO_2$ reduction.

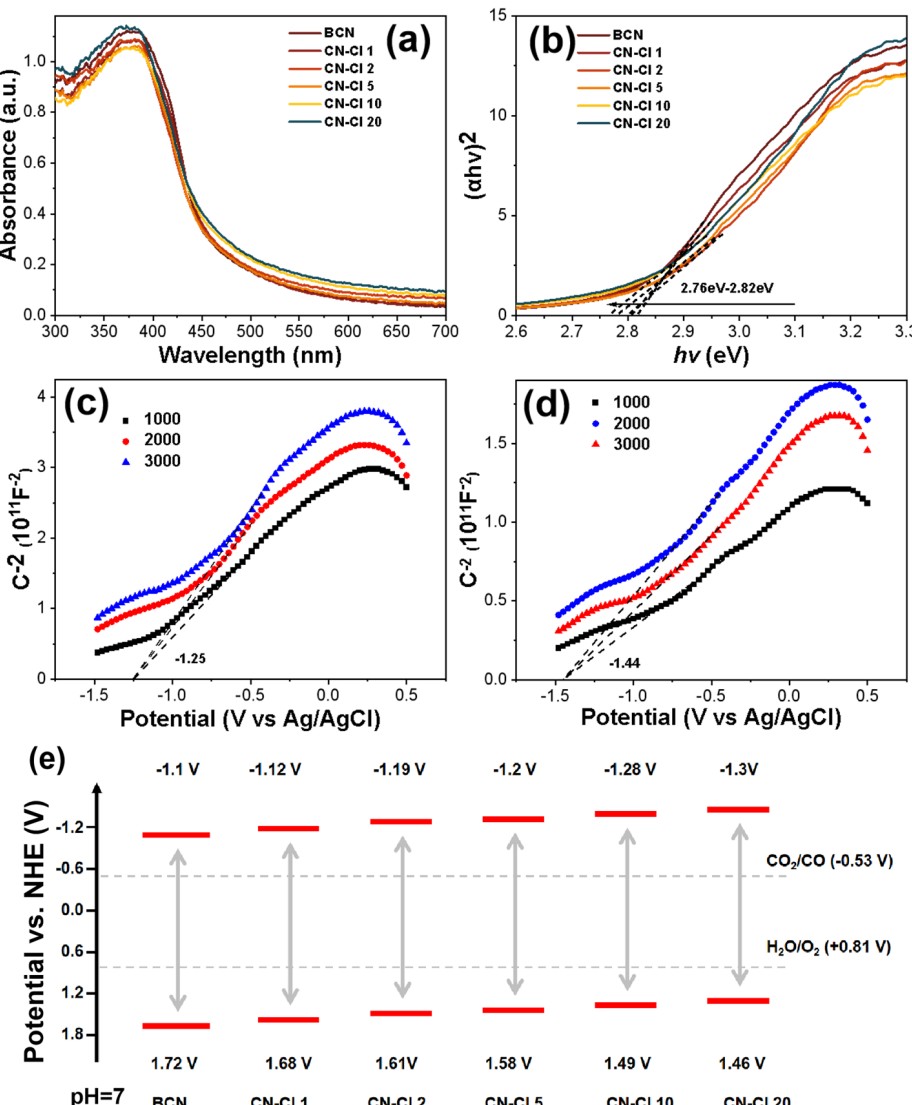

**Figure 7.** UV–visible diffuse reflectance spectra (**a**) and corresponding Tauc plots of the BCN and CN−Cl x samples (**b**). Mott–Schottky plots of the BCN (**c**) and CN−Cl 20 (**d**) samples, measured in 0.4 mol L$^{-1}$ of Na$_2$SO$_4$ solution (pH 5.95). Band diagram of the BCN and CN−Cl x samples, estimated from UV–vis spectra and Mott–Schottky plots (**e**).

Mott–Schottky plots were created to determine the flat-band positions of the photocatalyst. The slopes of the Mott–Schottky plots in the BCN and CN-Cl x samples are positive, indicating their typical n-type semiconductor characteristics (Figure 7c,d). The flat-band positions of BCN and CN-Cl 20 were determined as −1.25 V and −1.44 V (vs. Ag/AgCl, pH = 5.95), respectively. The CB potentials of BCN and CN-Cl 20 at pH = 7 were calculated as −1.1 V and −1.3 V based on Equations (5) and (6):

$$E_{Ag/AgCl} = E_{RHE} - 0.059\,pH - 0.197 \tag{5}$$

$$E_{NHE} = E_{RHE} - 0.059\,pH \tag{6}$$

Combined with the bandgap widths of the catalysts, the valence band positions of the BCN and CN-Cl 20 samples were determined as 1.72 eV and 1.46 eV, respectively [27]. The electronic band positions of the as-synthesized samples are illustrated in Figure 7e (a band diagram of the CN-Cl 20 sample is shown in Figure S2). It is clearly shown that Cl

modification of g-$C_3N_4$ can not only narrow the bandgap width, but also causes a negatively shifted CB potential level, which, therefore, enhances the photocatalytic reduction of $CO_2$.

In order to further investigate the photogenerated electron–hole separation efficiency of the catalyst, steady-state photoluminescence (PL) spectra were collected. The PL emission at 445 nm is assigned to the band–band recombination of photogenerated charge carriers (Figure 8a). The intensity of the peak at 445 nm in the CN-Cl x (x represents the amount of $NH_4Cl$) samples gradually decreases with the increase x, implying that introduction of Cl in g-$C_3N_4$ suppresses the recombination rate of photogenerated charge carries. CN-Cl 20 shows the lowest PL intensity, which accounts for its highest photocatalytic activity [28].

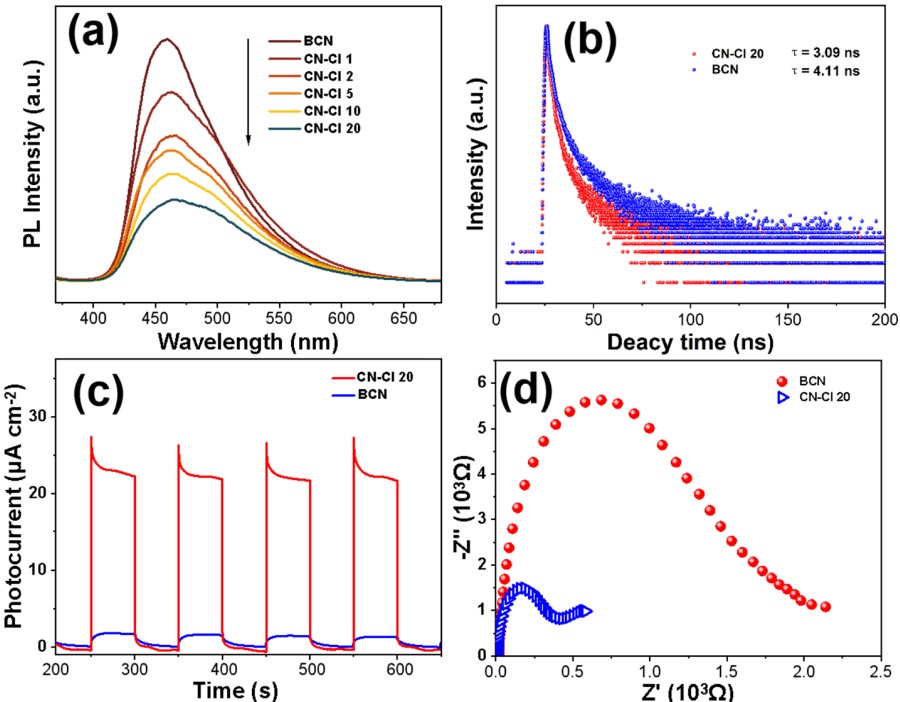

**Figure 8.** PL spectra of the BCN and CN−Cl x samples (**a**), time–resolved PL spectra (**b**), transient photocurrent response (**c**), and electrochemical impedance spectra (**d**) of the BCN and CN−Cl 20 samples.

To further investigate the charge transfer dynamic of the catalyst, transient photoluminescence was performed (Figure 8b). The lifetimes of BCN and CN-Cl 20 are 3.09 ns and 4.11 ns, respectively. The longer charge carrier lifetime means a longer diffusion length of the charge carriers, which is beneficial for charge separation and surface photoreaction.

In addition, from Figure 8c, we can see that the photocurrent of CN-Cl 20 is 10 times higher than that of BCN, indicating that doping CN with Cl can improve the interfacial charge transfer efficiency. Similarly, the electrochemical impedance spectrum (EIS) of CN-Cl 20 shows a smaller radius compared to BCN, also indicating that the introduction of Cl into CN can reduce the interfacial charge transfer resistance and stimulate interfacial charge separation (Figure 8d) [18]. These experimental results suggest the positive effect of Cl doping on the photoreactivity of CN.

### 3. Experimental Section

#### 3.1. Materials

All chemical reagents used in this study are analytical grade and were used without further purification. Dicyandiamide ($C_2H_4N_4$, AR), ammonium chloride ($NH_4Cl$, AR), absolute alcohol ($C_2H_6O$), and sodium bicarbonate ($NaHCO_3$, AR) were purchased from Sinopharm Chemical Reagent Co. (Shanghai, China) Deionized water was provided throughout the experiments.

### 3.2. Sample Synthesis

Bulk g-C$_3$N$_4$ (BCN) was prepared by the thermal polymerization method according to [22]. In detail, 4 g of dicyandiamide (DCDA) was calcined in the maffle furnace at 550 °C for 2 h in air. The ramping rate was set to 5 °C min$^{-1}$. The resulting yellow powder was collected after cooling down at room temperature. After that, the product was ground in an agate mortar and washed three times with anhydrous ethanol and deionized water, followed by filtering and drying at 60 °C for 16 h. For simplicity, the pristine g-C$_3$N$_4$ is denoted as BCN. The Cl-modified g-C$_3$N$_4$ catalysts were prepared via a gas template method. A mixture of 4 g of dicyandiamide and a certain amount of ammonium chloride (4, 8, 10, 20, and 80 g) was ground for 10 min. Consequently, the mixture was calcined at 550 °C in air at a ramping rate of 5 °C min$^{-1}$ for 2 h. The yellow product was collected after washing, filtration, and drying. According to the mass ratio of ammonium chloride to dicyandiamide (1:1, 2:1, 5:1, 10:1, 20:1), the obtained Cl-modified g-C$_3$N$_4$ was named as CN-Cl 1, CN-Cl 2, CN-Cl 5, CN-Cl 10, and CN-Cl 20, respectively.

### 3.3. Sample Characterization

The obtained samples were characterized by X-ray diffraction (XRD) patterns on a Rigaku D/MAX-RB diffractometer with Cu K$\alpha$ radiation ($\lambda$ = 1.5418 Å). X-ray photoelectron spectroscopy (XPS) measurements were carried out on a PerkinElmer PHI 5000C system, and all XPS spectra of the as-prepared samples were calibrated by setting the C1s peak of adventitious carbon to 284.6 eV. The morphology of the BCN and CN-Cl x samples was measured by transmission electron microscopy (TEM) (FEI talos f200s). The nitrogen adsorption–desorption isotherms of the obtained samples were recorded in a nitrogen adsorption apparatus (ASAP 2020, Norcross, GA, USA). The BET surface area (S$_{BET}$) of the samples was calculated by a multipoint BET method under a relative pressure range of 0.05–0.3. The corresponding pore size distribution curves were obtained by the Barret–Joyner–Halender (BJH) method. The UV-vis diffuse reflectance spectra (DRS) were collected on a UV-vis spectrophotometer (UV2550, Shimadzu, Osaka, Japan) with BaSO$_4$ as the reflectance. The steady-state photoluminescence (PL) measurement was carried out with a Fluorescence Spectrophotometer (F-7000, Hitachi, Tokyo, Japan) with an excitation wavelength of 350 nm. The electrochemical characterizations were tested in a standard three-electrode configuration (CHI760E, Shanghai, China).

### 3.4. Photocatalytic Reduction of CO$_2$

Photocatalytic CO$_2$ reduction was carried out using glassware (Porphyry). Firstly, 20 mg of the photocatalyst was dispersed into 15 mL of water to form suspensions. Secondly, the suspensions were transferred into a petri dish ($\Phi$ = 60 mm). After ultrasonication for 10 min, the dish was moved into an electric oven for drying overnight. The uniformly dispersed film was formed and placed in the reactor. Then, 1.2 g of sodium bicarbonate was added into the glassware. After that, the air was pumped out from the reactor. Next, 5 mL of sulfuric acid solution (2M) was injected into the reactor and reacted with sodium bicarbonate to produce carbon dioxide and water vapor in the system. After illumination under a Xenon arc lamp ($\lambda \geq$ 420 nm) for 1 h, 1.0 mL of the mixed gas was syringed and injected into a gas chromatograph (GC-2014, Shimazu, Osaka, Japan) for product analysis. The obtained gas chromatograph and detailed calculation process can be found in the Supplementary Materials.

When repeating the experiment, except for replacing fresh sodium bicarbonate and sulfuric acid, the other procedures were identical to the photocatalytic reaction mentioned above.

### 3.5. In Situ DRIFTs Measurement

In situ DRIFTs measurement for photocatalytic CO$_2$ reduction were performed on a FTIR spectrometer (Tensor II, Bruker, Bremen, Germany) equipped with a reaction chamber. Before measurement, the sample was pre-dried at 100 °C in a vacuum overnight. The dry

sample was subsequently placed in the sample chamber, followed by blowing with highly pure $N_2$ to replace air. After that, a mixture of $CO_2$ gas and $H_2O$ vapor was flowed into the sample cell for 60 min until the absorption equilibrium was achieved. The IR spectra of all BCN and CN-Cl x samples, both in the dark and under illumination, were recorded. A MCT detector was used during the measurement. The IR spectra were recorded by averaging 32 scans in the range of 600–4000 $cm^{-1}$ with a 2 $cm^{-1}$ resolution.

## 4. Conclusions

A highly active g-$C_3N_4$ photocatalyst towards $CO_2$ reduction was prepared by direct calcination with the mixture of dicyandiamide and ammonium chloride. The introduction of Cl into g-$C_3N_4$ changed its band alignment, resulting in a narrower bandgap and a negatively shifted CB position. The ammonia gas, from the decomposition of ammonium chloride, acted as a gas template to exfoliate g-$C_3N_4$, which can improve the specific surface area. The CN-Cl 20 sample showed 3.1 times higher photocatalytic activity of $CO_2$ reduction than that of bulk g-$C_3N_4$ due to the reduced bandgap, negatively shifted CB potential position, and enlarged specific surface area. This work provides a facile way to prepare a highly efficient g-$C_3N_4$-based photocatalyst for $CO_2$ conversion and utilization.

**Supplementary Materials:** The following supporting information can be downloaded at: https://www.mdpi.com/article/10.3390/catal13091260/s1. Figure S1. XRD patterns of the as-prepared CN samples in diffraction angles of 26° to 30°; Figure S2. Electronic band structures of NCB and CN-Cl 20 sample; Table S1. Yields of the products for the synthesis of carbon nitride photocatalysts; Figure S3. Working curves for CO (A) and $CH_4$ (B), respectively; Figure S4. Gas chromatograph of photocatalytic $CO_2$ reduction; Table S2. Comparison of the reaction rate for photocatalytic $CO_2$ reduction.

**Author Contributions:** Conceptualization, K.L.; methodology, K.L.; formal analysis, Z.W., S.C., M.Y., Z.Z., Q.L. and K.L.; investigation, Z.W.; resources, K.L.; data curation, Z.W., S.C., M.Y. and K.L.; writing—original draft preparation, Z.W. and K.L.; visualization, Z.W.; writing—review and editing, Z.W. and K.L.; supervision, project administration, and funding acquisition, K.L. All authors have read and agreed to the published version of the manuscript.

**Funding:** This work was financially supported by the National Natural Science Foundation of China (51672312, 21972171 and 22305132), the Hubei Provincial Natural Science Foundation of Huangshi, China (2022CFD001), and the Fundamental Research Funds for the Central Universities, South-Central Minzu University (CZP22001, KTZ20043 and CZZ21012).

**Data Availability Statement:** Not applicable.

**Conflicts of Interest:** The authors declare no conflict of interest.

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
