# Peer review of "Boosting the Photoreactivity of g-C3N4 towards CO2 Reduction by Polymerization of Dicyandiamide in Ammonium Chloride"

_catalysts, doi:10.3390/catal13091260_

Round 1

Reviewer 1 Report

There are some problems with the content which must to be clear the scientific objects. I suggest the status of “rejection” for the manuscript and major revision from the authors is required before the further reviewing.

The English in the present manuscript is not of publication quality and require major improvement. 

Reviewer 2 Report

In this article, Zhi Wang et.al, demonstrated the synthesis of Cl-GCN-based photocatalyst for CO2 reduction. The authors have explored the technology that can optimize the properties of the materials used in photocatalytic CO2 reduction with the help of In-situ FTIR analysis. The prosed mechanism and results are satisfactory. The manuscript was well designed, and the results are interesting, and the manuscript organization is good. However, there are a few aspects that authors should focus on for enhancing the quality of the manuscript as noted below, before it can be published.

(1)   It is necessary to explain abbreviations when they first appear, such as (CO2RR).

(2)   Please remodify the sentence in line NO: 39 to 41 on page NO:1. After this sentence, please explain the g-C3N4 significance among 2D materials along with their drawbacks in more detailed with proper citations.

(3)   It is much useful for the readers if the authors can explain the mechanism of the formation of CN-Cl 20 formation from the NH4Cl and how it improves the surface area, does it cause the Cl as a dopant or substitutional intercalation with the CN matrix. Provide a suitable reference. Please insert the Zoomed view of Figure 2B for a better view of small-sized pore.

(4)On page 2, from line NO: 70 to 76, the explanation about the controlled use of NH4Cl amount in the experiment is a little bit confusing. Please improve it. And the sentence grammar needs to be improved. Please check there are some grammatical errors.

(5)   Have the authors checked the Photocatalytic CO2 reduction performance of samples above the 1:20 ratio.

(6)   It could be much better if authors can elaborate on the mechanism of CO conversion in more detail rather than the characterization techniques. But it was very by portraying the phenomenal mechanism using in-situ FTIR; however, using band structure or pictographic representation gains much attraction. Please consider it; if not ignore it.

(7)  How about long-term stability?

(8) This work investigated the Cl-based GCN-enabled photocatalytic CO2 reduction. Some relative papers may enrich the concepts and background of this work as references: The Journal of Physical Chemistry C 125.18 (2021): 9646-9656 , Nano Research 16.5 (2023): 7682-7695, Physicochemical and Engineering Aspects 611 (2021): 125780. ACS Applied Materials & Interfaces 15.15 (2023): 18898-18906, 

Minor sentence errors. It can be verified if the authors can proofread the entire manuscript.

Reviewer 3 Report

This manuscript systematically studied the fabrication of g-C3N4 photocatalytic material by polymerization of dicyandiamide (C2H4N4) in the presence of ammonium chloride (NH4Cl), which exhibited enhanced photoreactivity towards CO2 reduction due to the combined effects of negatively shifted CB potential and enlarged BET surface area. In situ FTIR spectrum was used to study the photocatalytic CO2 reduction mechanism. This manuscript is publishable. However, the quality of the manuscript can be further improved by considering the following questions.

1. How about the effect of ammonium chloride amount on g-C3N4 yield?

2. From Fig. 6a, it can be seen that the photoreactivity of g-C3N4 steady increases with increasing the amount of ammonium chloride. Why not further increase the amount of ammonium chloride to optimize the photocatalytic activity of g-C3N4?

3. SEM images of BCN and optimal photocatalyst (CN-Cl 20) can be presented and compared.

4. Both CO and CH4 are main products of CO2 reduction. However, in the present study, I can not find the result of CH4 production.

Round 2

Reviewer 1 Report

 Accept in present form

Reviewer 2 Report

Accept

Reviewer 3 Report

It is well revised and can be accepted now.